

# Assessment of tropospheric ozone products from CAMS reanalysis and near-real time analysis using observations over Iran

Najmeh Kaffashzadeh[1] and Abbas Ali Aliakbari Bidokhti[1]

[1] Institute of Geophysics, University of Tehran, Tehran, Iran

*Correspondence to*: Kaffashzadeh Najmeh (n.kaffashzadeh@ut.ac.ir)

**Abstract.** Tropospheric ozone time series consists of the effects of various scales of motion, from meso to large time scales which is often challenging for global models to capture them. This study uses two global datasets, namely reanalysis and analysis of the Copernicus atmospheric Monitoring Service (CAMS), to assess the capability of these models or systems in presenting ozone's features in small scales. We employ the tropospheric ozone product of the models and in situ measurements

at 18 stations over Iran for the year of 2020. Furthermore, we make use of data of ozone, temperature, nitrogen oxides, wind speed, and wind direction at one more station. We decompose the datasets into three spectral components, i.e., short (S), medium (M), and long (L) term. We only evaluate the S and M terms of modelled against those of observed datasets for all stations. We examine the relationship between ozone and the relevant proxies. Results show a correlation coefficient of larger than 0.5 for S and about 0.25 for M term in both models. It turns out that the reanalysis dataset demonstrates more precision

for the S component than that for the M. Both models can show the observed correlation between ozone and temperature, whereas some inconsistencies appear in presenting the anti-correlation between ozone and nitrogen oxides.

## 1 Introduction

Near surface ozone ($O_3$), or tropospheric ozone at the ground level, is a secondary air pollutant that deteriorates human health and plants via damaging respiratory systems (Bell et al., 2006; Fowler et al., 2009; Mills et al., 2011; Malley et al., 2015).

Surface ozone is either transported naturally from the stratosphere or produced in situ by photochemical oxidation of ozone's precursor gases such as nitrogen oxides, non-methane volatile organic compounds, methane or carbon monoxide in the presence of sunlight (Crutzen 1974; Monks et al., 2015; Cooper et al., 2014). The ozone level is not only a function of its precursor's emissions, but also meteorological conditions that influence the evolution of emissions and photochemical products (Bloomer et al., 2009; Li et al., 2020).

Global chemistry climate models, as a tool that the user can rely on, provide an estimation of the radiative forcing of tropospheric ozone from preindustrial period to the present and project this variation into the future in a global scale (Stevenson et al., 2006; Murazaki and Hess, 2006; Lin et al., 2008a). These models have been widely used as an initial condition for the daily forecast of the atmosphere or boundary conditions in regional models, and as proxies to complement insufficient in situ measurements. Often these models contain coarse grid boxes, i.e., 100 km, which prevent an adequate representation of





mesoscale meteorology cyclone wave activity that leads to uncertainty in ozone precursor's transportation (Lin et al., 2008b). The mesoscale events such as land-sea breezes and topographically forced flows determine whether ozone precursor species remain locally or transport downwind. Representation of mesoscale circulations associated with cyclones could be achieved with a resolution less than 0.25°.

Iran is a country with a complex topography and diverse meteorological systems as it extends from about 25° N to nearly 40° N latitude. Many efforts have been done to study ozone and its precursors over this country (Lelieveld et al., 2009; Bidokhti et al., 2016; Faridi et al., 2018; Yousefian et al., 2020). Most of the studies have been conducted over Tehran megacity, which suffers from severe ambient air pollutions (Zohdirad et al., 2020; Borhani et al., 2021; Jafari Hombari and Pazhoh, 2022). They have shown that not only local emissions and winds, but also synoptic conditions control the ozone levels. Synoptic systems, which cause to high level of ozone, over Tehran has been classified in a study by Khansalari et al. (2020) and Lashkari et al. (2020). Taking into account of the complex topography of the country and the contribution of various factors in different time scales on ozone: can we use global models data to study the surface ozone over Iran? Or which global models are (more) suitable for the study of ozone over this country?

To answer these questions, we select two types of global models' data. The first is a global model that is assimilated by various observations such as satellite, radar, and in situ measurements, that is called reanalysis. The latter is a global model with a fine resolution and the large number of vertical levels that can resolve the orography. We use measured ozone data at 19 stations for the year of 2020, in which the second model data has been available with the required resolution. To assess ozone in each time scales, i.e., meso, synoptic or large scale, the observed and modelled datasets are decomposed to three spectral components. Then the models' performances are evaluated and compared for each spectral component. Relationships between ozone data and a few relevant proxies are quantified for the observations and models datasets that identify the models' capability in representing these relationships.

A detail description of the datasets (observations and models) used in this study is presented in Sect. 2. Methodology is explained in Sect. 3 and the results are shown in Sect. 4. Discussion is presented in Sect. 5 and the paper ends with conclusion' remarks in Sect. 6.

## 2 Description of Data

### 2.1 Models

#### 2.1.1 CAMS reanalysis

The Copernicus Atmosphere Monitoring Service (CAMS) has been developed in recent years to assimilate the chemical compositions such as tropospheric ozone and aerosol concentrations, but it also holds outputs for several meteorological variables (Innes et al., 2019). The CAMS product is the latest (state-of-the-art) global reanalysis datasets of atmospheric compositions. It is noteworthy that newer versions of data have been frequently adopted in CAMS. Comparing to the previous



atmospheric chemistry reanalysis data, CAMS has a finer horizontal resolution of 80 km with 60 vertical levels. CAMS covers data for the period of the January 2003 to June 2020. The data are archived in 3 hourly time intervals. Several studies have evaluated CAMS products and compared them with other reanalysis datasets and control run (no assimilation). As an example, an inter-comparison of tropospheric ozone from seven reanalysis datasets in East Asia has reported that CAMS depicts more

reasonable spatial-temporal variability than other datasets (Park et al., 2020). They also show the suitability of CAMS for the study of local tropospheric ozone on seasonal to interannual time scales, but inadequacy of that to study trend (long-term). Despite the well performance of CAMS in upper troposphere, it shows bias at the surface for many parts of the globe (Wang et al., 2020). Meanwhile, several of these studies mention that the performance of CAMS differs depending on the region. For instance, it has been shown more agreement between CAMS and observations over Europe than Tropics (Errera et al., 2021).

**2.1.2 CAMS near real-time analysis**

In addition to aforementioned datasets, CAMS provides global (near real-time) forecasts, called analysis, of various chemical compositions and meteorological variables twice a day. In contrast to the reanalysis datasets, the forecast data are available on a finer resolution of 40 km. This system is upgraded regularly, e.g., once a year, in which model's resolution can change or new species can be added. On July 2019 the model vertical levels have been upgraded from 60 to 137. The temporal coverage

of the data is from 2015 to present with temporal resolution of 1 hourly and 3 hourly (within five forecasts days). System's upgrades and its verification are reported in several studies (Shulz et al., 2021; Eskes et al., 2021). A recent validation based on various observations shows that, in terms of bias, analysis overestimates surface ozone values at most of the stations (Sudarchikova et al., 2021). However, it shows a significant correlation over most of the stations, e.g., in China.

**2.2 In situ measurement datasets**

Surface based measurements of ozone were extracted from Tehran air quality control portal for 19 stations. Hourly time series of surface ozone were obtained from Iran Environmental Protection organization for eight additional stations. We add Geophysics station, which contains not only data for ozone but also for a few proxies such as air temperature, nitrogen oxides, and wind. To have a common quality, validity of the data was checked by performing a few statistical tests such as range, constant values, and missing value tests suggested in literatures (Zurbenko et al., 1996; Zahumensky, 2004; Gerharz et al.,

2011; Taylor and Loescher, 2013). The invalid values were majorly flagged in the missing values test, which identifies suspicious data points before and ahead of the discontinuities. We use the stations containing data for the year of 2020, where more than 50 percent of the data are available for each month. Table A1 lists the names and geographical locations of the stations, which are ordered based on the stations' latitudes. In this table, there is a number along with the stations' name, hereafter, the stations are referred with these numbers.

Hereafter, the observation, reanalysis and analysis datasets are called as dfo, dfr, and dfa, respectively.





## 3 Methodology

Observed data are available in hourly resolution, in contrast to the model values that are available in 3 hourly intervals. To match the frequency of the model outputs with observations, 3 hourly observed values are considered in such a way that at least two hourly values are available, otherwise it renders the value as a missing data. The vertical model level at each station
location was selected based on the minimum difference between the station altitude and the central altitude of the model level obtained when converting model level pressure to altitude with the barometric equation and using a scale height of 7640 m. For surface pressure, the annual average surface pressure of the grid box was used.

### 3.1 Spectral decomposition of the time series

The presence of various scales of motions, which caused by several physical and chemical processes, in time series of $O_3$ can
complicate analysis and interpretation of data. As an example, short-term and fast fluctuations in $O_3$ time series are majorly attributed to the chemical process such as NO titration, whereas long-term and seasonal variation is mainly related to the solar radiation. Scale analysis is a method by which the time series can separate into different temporal terms. Here, the time series of $O_3$ is decomposed into three different spectral components, namely short (period less than 2 days), medium (period of 2-21 days), and long (periods longer than 21 days) terms by applying Kolmogorov-Zurbenko (KZ) technique (Rao et al., 1997). KZ
is essentially a low pass filter, which consists of repeated moving average. Its use has been demonstrated in earlier studies (Hogrefe et al., 2000, 2014; Kang et al., 2013; Seo et al., 2014). A detailed discussion of the KZ filter along with the comparison to other separation techniques can be found in Eskridge et al. (1998) and Loneck and Zurbenko (2020). KZ requires two input parameters, KZ (m, k), where m is the window size for filtering and k is the number of iterations. Since the values that have been commonly used for m and k in the literature may not be applicable for 3 hourly data, we selected them based on the
criterion suggested in Yang and Zurbenko (2010):

$$m \times \sqrt{k} \leq p \qquad (1)$$

KZ filters out all periods that are less than p. Therefore, three components of interest in this study are estimated as following:

$$S = O - KZ\ (5,\ 5) \qquad (2)$$

$$M = KZ\ (5,\ 5) - KZ\ (35,\ 5) \qquad (3)$$

$$L = KZ\ (35,\ 5) \qquad (4)$$

where O refers to the original time series and S, M, and L indicate the short, medium, and long terms, respectively.

### 3.2 Model evaluation

We use the mean square error (MSE) as a metrics to evaluate the models' performance. The MSE is defined as the squared mean of the difference between modelled ($x_m$) and observed ($x_o$) variables.



This metric can be modified to include all relevant model evaluation indicators, i.e., bias, variance, and correlation, as (Murphy, 1988; Solazzo and Galmarini, 2016):

$$MSE = (\bar{x}_m - \bar{x}_o)^2 + (\sigma_m - r\,\sigma_o)^2 + \sigma_o^2(1 - r^2) \tag{5}$$

where $\sigma_m$ and $\sigma_o$ refer to the standard deviation of the modelled and observed data respectively, and r is the coefficient of correlation between the observed and assimilated datasets. In Eq. (5), the first term (hereafter $e_1$) shows the deviation between average modelled ($\bar{x}_m$) and measured ($\bar{x}_o$) datasets and refers to the model accuracy. The second term (hereafter $e_2$) contains the variance error, i.e., the discrepancy in amplitude or phase between the variability of the modelled and observed values, that determines the precision of the model. Also, the third part (hereafter $e_3$) refers to unsystematic errors related to the associativity between observed and assimilated datasets. In other words, the $e_2$ indicates an explained error which reveals the variance error arising from the variability of the modelled variables not observed in measurements. The $e_3$ represents an unexplained error reflecting the lack of observed variability in the modelled data. Due to the spectral decomposition of the data, the S and M components are zero-mean fluctuations. Hence, the $e_1$ term in Eq. (5) is zero and only the $e_2$ and $e_3$ terms are analysed below.

### 3.3 Multiple linear regressions

To find a quantitative relationship between variability of $O_3$ and several predictors, a multiple linear regression (MLR) is defined as:

$$O_3(t) \approx a_0 + a_1\,NO_x(t) + a_2\,AT(t) + a_3\,WS(t) + a_4\,WD(t) \tag{6}$$

where $a_0$ is an intercept, $a_1$, $a_2$, $a_3$, and $a_4$ are the regression coefficients for $NO_x$ (nitrogen oxides), AT (temperature), WS (wind speed), and WD (wind direction). These predictors are relevant proxies for the regression model of $O_3$ (Bloomfield et al., 1996; Abdul-Wahab et al., 2005; Rasmussen et al., 2012; Otero et al., 2018). In order to allow comparison between the regression coefficients of the different variables, the MLR is performed over standardized data, i.e., mean 0 and standard deviation of 1, for both S and M components.

### 4 Results

The time series of $O_3$ and all meteorological variables for observations and models decompose into three spectral, short (S), medium (M), and long (L), by applying the method (KZ filter) explained in Sect. 3.1.

Figure 1 shows the original $O_3$ time series of observations and two models and its estimated spectral components at first station. To clearly see the signals, we only show part of the time series, here the summer month (June, July, and August: JJA). Looking at the original 3 hourly times series (a), both models overestimate and underestimate ozone during different periods, but it is difficult to determine any clear patterns or identify specific reasons for the model bias. The S component contains frequent fast oscillations occurring every day with regular maxima and minima (see panel (b) in Fig. 1). In this figure, the amplitude of S oscillations in the two models is different from that in dfo, indicating differences in the diurnal cycle of observed and simulated ozone mixing ratios. The M term captures variability on the time scale of synoptic systems. Some episodic events





are more visible in the M component than in S. For instance, in panel (c) of Fig. 1, the M component of the observation represents a clear signal of an episodic event in middle of June. This episode is not well captured in dfr while they are captured by dfa. It seems for most of the periods the variations of M component in both models are in good agreement with those in observations while the amplitudes of oscillations in the models do not correspond well with that of the observations. The

underestimation and overestimation of the amplitude (with respect to observations) in dfa is less than that in dfr. Compared to S and M terms, which oscillate around zero, the mean values of the L components are not zero (see panel (d) in Fig. 1). L represents variations of the ozone mixing ratios on seasonal, semi-seasonal, and multi-annual time scales. Comparing the variations of dfr and dfa with dfo for L shows more similarity between dfa and dfo than between dfr and dfo. Both models exhibit a high bias with respect to the ozone mixing ratios. Nevertheless, the decomposition of L component is not reliable due

to limited period (one year) of the available data, so hereafter we only assess the S and M components.

Figure 2 shows the box plots of MSE and different terms of MSE, i.e., $e_2$, $e_3$, for $O_3$ simulation in both components. From panel (a) in Fig. 2, it turns out the mean MSE (shown with red squares) of $O_3$ for S component is larger than for M component for both models. In other words, there is a better agreement between M components of observations and models than S components. That arises from larger variability of S than M (see Fig. 1). The mean of MSE in dfr for S component (dfr_s) is

larger than that in dfa (dfa_s). Whereas, for the M component the differences between MSE of dfr and dfa is less. So, S consists of more variability in dfr than that in dfa. Panel (b) in Fig. 2 shows the explained error ($e_2$) in dfr and dfa for both components. The higher $e_2$ for S in comparison to that for M component can contribute to the large MSE of S. Since $e_2$ is a model related error, a possible source for the large $e_2$ of S can be the misrepresentation of short and meso scale phenomena in models. As can be seen in the model with a higher resolution, i.e., dfa, this error is less than that in dfr. This error could also arise from

the assimilated processes which causes a larger variability in dfr than dfa. The unexplained error ($e_3$) for both components is presented in panel (c) of Fig. 2. Similar to $e_2$, the $e_3$ for S component is larger than that for M as expected from the variance of these components. Large value of $e_3$ for S component can arise from the models' deficiency in resolving the meso-scale phenomena such as local winds, NO titration and their influences on $O_3$ variability.

Assessing the element of $e_2$ (see second term of Eq. (5)) shows that the large variance of the model ($\sigma_m$), small variance of

observations ($\sigma_o$), or small correlation (r) causes the large $e_2$ and consequently large MSE. Panel (a) of Fig. A1 shows the correlation between the models and observation datasets for both components. This figure shows that the S contains a larger correlation (r > 0.5) than M (r ≈ 0.25) in both models. In addition, the mean value of correlation ($\bar{r}$) for the S component of dfr (= 0.62) is larger than that for dfa (= 0.51), while for the M of dfr and dfa it ($\bar{r}$) is 0.19 and 0.22, respectively. A high value of correlation between two terms can be attributed to the larger covariance of two terms or less variance of each term. panel

(b) of Fig. A1 shows the covariance between models and observations. As can be seen in this figure, the covariance between dfr and dfo (= 407.01) is much larger than that for dfa (= 97.5) for S component, while that is nearly the same for M. The S of dfr (dfr_s) shows larger variance than that of dfa (dfa_s), whereas its M component (dfr_m) contains less value than dfa (dfa_m) in panel (c) of Fig. A1. In this figure, the mean value of variance for dfa_s and df_m is close to that for dfo_s and





dfr_m, respectively. So, the high correlation is majorly attributed to the large covariance, and large value of $e_2$ is caused by the

models' variance.

Four stations, i.e., 1, 2, 3, and 15, appear as fliers (see black circles in panel (a) of Fig. 2) with low value of MSE in dfr, less than 500 and 20 for S (dfr_s) and M (dfr_m), respectively. That arises from the low value of $e_2$ (see panel (b) in Fig. 2). A smaller correlation (and covariance) between simulated and observed $O_3$ are shown at these sites comparing to other stations (see circles for dfr_s in panel (a-b) of Fig. A1). Nevertheless, at these sites, the variances of dfr_s and df_m is less than other

stations (panel (c) of Fig. A1). Thus, the small value of MSE at these stations is mainly attributed to the low model variabilities. The variabilities of the spectral components of the observed $O_3$, with respect to the relevant proxies, are quantified by using a MLR model explained in Sect. 3.3. Table 1 lists the regression coefficients $a_i$ ($i = 1...4$) and coefficient of determination ($R^2$) for both components of observations at the station 8. Results of the regression coefficients in this table indicate a small difference between S and M for the regression coefficients of $a_3$ and $a_4$, however there is a large difference between $a_1$ and $a_2$,

as predictors in S and those in M. For the S component, $a_1$ is -0.39 which means upon increasing $NO_x$ by 1 unit (= 43.36 nmol mol$^{-1}$, see table A2), $O_3$ decreases by 0.39 units (= 18.11 nmol mol$^{-1}$, see table A2), holding other parameters constant. However, one unit change in M component of $NO_x$ leads to a reduction of 0.34 unit of $O_3$ (= 4.88 nmol mol$^{-1}$, see table A3). Highest coefficient belongs to the AT ($a_2$ = 0.63) of the S component and indicates that the large association between short variabilities of AT and that of $O_3$. This association decreases to 0.23 for M. From the results in this table, it appears that the $R^2$

for the S is 0.67, while that is 0.2 for the M. This shows that the MLR model explains more variability of S than M.

Figure 3 shows regression coefficients and $R^2$ values for both S (a) and M (b) components of dfo, dfr, and dfa at station 8. While some coefficients span both positive and negative, $a_2$ is always positive (correlation). For instance, $a_2$ for S component of dfr and dfa are 0.71 and 0.81, which means upon increasing AT by 1 unit (= 2.69 °K of dfr and 2.88 °K of dfa, see table A2), $O_3$ increases by 0.71 units (= 31.9 nmol mol$^{-1}$, see table A2) in dfr and increase 0.81 units (= 15.46 nmol mol$^{-1}$, see table

A2) in dfa. Although both dfr and dfa are in a good agreement with observations in presenting the positive correlation between $O_3$ and AT (see Fig. 3), the behavior of other predictors are highly variables. Similar to dfo, there is an anti-correlation between S component of $O_3$ and $NO_x$ in dfr, while dfa shows a positive correlation. On the other hands, for M component dfr shows a correlation between $O_3$ and $NO_x$, whereas dfo and dfa show an anti-correlation. The negative association (anti-correlation) between $O_3$ and $NO_x$ presented in dfo, is captured by S of dfr and M of dfa. In addition to the sign of coefficients in this figure,

its absolute value is highly variable compared to $a_2$. It is clear that for dfo where $|a_1|$ value is high (= 0.39), it falls to 0.11 for dfr and to 0.15 for dfa. This shows a smaller contribution of $NO_x$ to $O_3$ in models comparing to that of observations. $R^2$ for S component of dfr and dfa are 0.68 and 0.56, respectively, while for M that is 0.23 and 0.09. Smaller $R^2$ of dfa comparing to that of dfo indicates that the model parametrization yields a weaker relationship between predictors and $O_3$ than that in observations.



## 5 Discussion

Analysis of the spectral components in this study reveals that the O$_3$ variabilities in both models possess a nearly similar shape (although in different phase and amplitude) as those in observations. Results of the models' performances show a larger MSE of S than that of M in both models. That arises from the larger variabilities of S in comparison to M (Hogrefe et al., 2000; Hogrefe et al., 2014). The results of error apportionment show that the MSE majorly has been arisen from the e$_2$, which emphasizes the modeling related error such as large models' variance. The e$_2$ assessment shows more variability for both component of O$_3$ in models than that in observations. That could arise from the complex chemical processes in the models or imbalanced among coupled components (Park et al., 2020). Large variance of dfr with respect to dfa and dfo can be attributed to the assimilation's procedures.

Larger correlation coefficient of S comparing to that of M reflects a better agreement between short-term variabilities of the models and observations than the synoptic term. That is attributed to the models' capability in simulating diurnal (24 hourly) cycles as has been shown for the dfr (Huijnen et al., 2020). Model to model differences in correlation coefficient is more pronounced in S, as the correlation of dfr is slightly larger than dfa. That can be related to the assimilation of ozone's precursors such as NO$_2$, which affects the O$_3$ in the short time scale.

At four stations, dfr shows less correlation with respect to other stations. Three of these stations are in city of Tabriz, in north western part of Iran, where is often affected from cyclonic activities (Asakereh and Khojasteh, 2021) and summer circulations over eastern Mediterranean region (Tyrlis et al., 2013). Although CAMS reanalysis captures the long-range transport processes and atmospheric background in the troposphere, it shows a lower skill over Mediterranean, in particular eastern part, compared to other regions (Errera et al., 2021).

Large MSE at some stations could arise from the selected vertical level. Figure A2 shows the O$_3$ time series for dfo and two vertical levels of dfa at station 4. The time series of dfo contains lots of fast daily fluctuations, which are not pronounced at the selected vertical model levels of dfa (lev = 123) as those in observations. Whereas, at the surface level (lev = 137) there is more matching between oscillations of dfa and that of dfo. To assess the sensitivity of results to the vertical level, the results of model performance was obtained for dfa when they are interpolated at the model level of 137 (Fig. A3 panel). At the new vertical model levels, the mean of MSE decreases from 202.07 to 162.34 for S component and from 42.45 to 21.1 for M component of dfa. In this case, similar to the results in Sect. 4, the MSE of S component is larger than that for M.

A positive correlation between O$_3$ and AT is shown for both components of all datasets. That is consistent with the results of other literatures (Ordóñez et al. 2005; Camalier et al., 2007; Otero et al., 2016; Otero et al., 2018) which show that the higher AT leads to higher O$_3$ through different processes. High AT leads to high photolysis rate and high concentration of O$_3$ precursors from dissociation of PAN, biogenic emission of isoprene, etc. The S component of observations shows the most significant correlation between O$_3$ and AT, which is attributed to the diurnal variation. Although both models capture the true relationship (correlation) between O$_3$ and AT with slight underestimation, the anti-correlation between O$_3$ and NO$_x$ are not well captured by them.





As an example, the S component of dfa and the M component of dfr show a positive correlation between $O_3$ and $NO_x$, despite the anti-correlation between observed $O_3$ and $NO_x$. In the study by Wang et al. (2020), it has been shown that the peak values

of some species such as NOx and CO is underestimated in dfr. Wind can influence $O_3$ concentration near the surface in a different way. If the site acts as a (point) source of $O_3$, the increase of the wind speed disperses (dissolute) ozone more efficiently from that place. Conversely, if the ozone chemical budget at one station is negative, wind can transport (transfer) the ozone-rich air from polluted regions to that site; in this case $O_3$ correlate positively with the wind speed. Nevertheless, in this study, the low value of $a_3$ and $a_4$ might arise form a non-linear relationship between $O_3$ and winds or unresolved time

scales; mountainous area of Iran often leads to local mesoscale circulations that can influence these processes and change near surface ozone, especially in urban areas as Tabriz or Tehran (Soltanzadeh et al., 2011). A weak (insignificant) correlation between wind speed and ozone level has been reported in other studies (Dawson et al., 2007; Otero et al., 2018). Although the selected parameters in MLR explained 67 % variabilities of S, they could explain only 20 % of the M. This indicates the role of other parameters that are not included in the MLR. For instance, Kaffashzadeh (2018) showed that the relative humidity is

the most influential factor on the synoptic surface ozone variability over Mediterranean region.

**6 Conclusions**

In this paper, the variability of $O_3$ in two global models, namely reanalysis and analysis, was assessed against that in observations at 19 stations over Iran. We decomposed the $O_3$ times series of all datasets to three spectral components, i.e., short (S), medium (M) and long (L) terms. The S contains intraday and diurnal fluctuations, the M term include synoptic fluctuations

and the other fluctuation, i.e., seasonal, semi-seasonal, and trend, carries in L. We only assessed the S and M terms due to the availability of one year data, i.e., 2020; the L component is primarily used to check the biases between model data and observations, but should not be considered reliable with respect to trend analysis etc. Since S and M components have zero-mean fluctuations, the bias term (distance between time average of model data and observations) is zero, and the main focus of this study was to analyse the variability term, e.g., variance and covariance. To assess how some processes involved in the

observed $O_3$ variability, we make use of a few relevant proxies such as $NO_x$, AT, WD and WS as predictors in the MLR model. The results presented in this study reveal three key points: (1) the S component of both models shows larger correlation with observations than M term. This indicates a higher capability of models in simulating the S variability of ozone compared to the synoptic term. (2) To study the S variability of $O_3$, one should use the assimilated model data, which shows a high correlation with observed data. (3) Despite presenting the correlation between AT and $O_3$, the anti correlation between $NO_x$

and $O_3$ are not well captured in both models. That can be considered as a starting point to improve the results of tropospheric ozone, in particular at urban sites.

**Code availability.** The Python 3.7 code of the methodology will made be available to reader under Creative Common license on the GitHub repository of the corresponding author.





**Data availability.** Part of observational data are accessible from _ portal. CAMS reanalysis and analysis data was obtained
through ECMWF's climate data service (last access April 2022).

**Author contribution:** N. K designed the research, acquired and processed all data, performed the statistical analysis and
composed the figures and manuscript. A. A. B. contributed to proofreading and discussion.

**Competing interests:** The authors declare no competing interests.

**Disclaimer:** Publisher's note: Copernicus Publications remains neutral with regard to jurisdictional claims in published maps
and institutional affiliations.

**Acknowledgements.** We thank the data providers of Iran Environmental Protection organization, Tehran Air Quality Control
Company, and the ECMWF's climate data services.

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

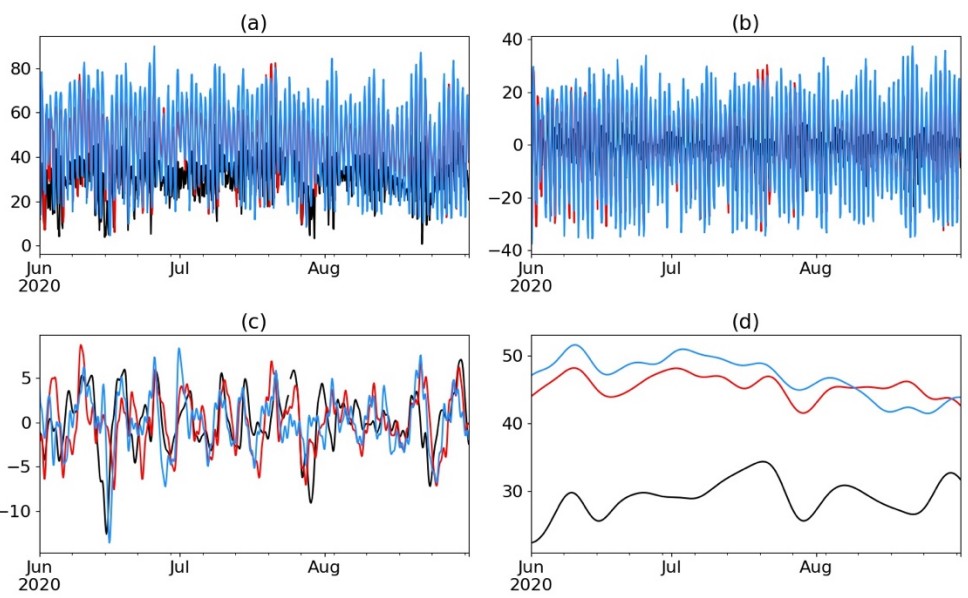

**Figure 1. It shows different spectral components of O$_3$ for dfo (black), dfr (red) and dfa (blue) at station 1 for JJA. (a) presents the original time series and (b), (c) and (d) are short (S), medium (M) and long range (L) fluctuation terms, respectively. The vertical axis in all panels show ozone mixing ratio in nmol mol$^{-1}$.**


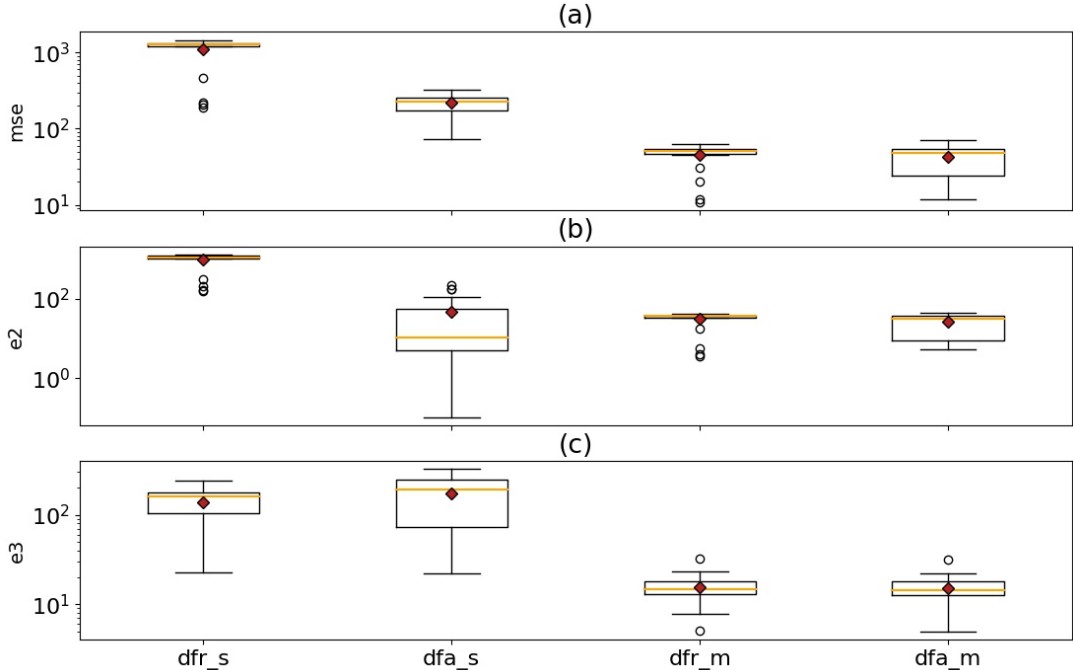

**Figure 2. This figure shows the (a) MSE, (b) e$_2$, and (c) e$_3$ of dfr, and dfa for both S and M components of O$_3$. In this figure, dfr_s and dfa_s refer to the S component of dfr and dfa, respectively. Likewise, dfr_m and dfa_m refer to the M component of dfr and dfa, respectively.**






**Table 1. The regression coefficients, i.e., $a_1$, $a_2$, $a_3$, and $a_4$, and coefficients of determination, i.e., $R^2$, for S and M components of dfo dataset at station 8.**

| component | S | | M | |
|:---:|:---:|:---:|:---:|:---:|
| $a_1$ | -0.39 | 0.01 | -0.34 | 0.02 |
| $a_2$ | 0.63 | 0.02 | 0.23 | 0.02 |
| $a_3$ | -0.08 | 0.01 | 0.08 | 0.02 |
| $a_4$ | 0.08 | 0.01 | 0.01 | 0.02 |
| $R^2$ | 0.67 | | 0.20 | |



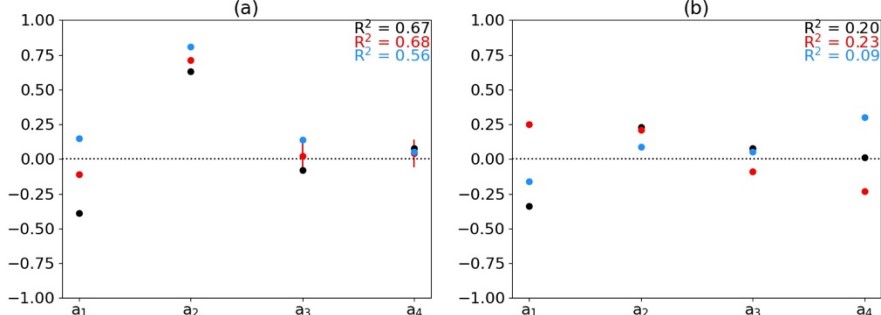

**Figure 3. This figure shows regression coefficient of (a) S and (b) M components for dfo (black), dfr (red) and dfa (blue) datasets at station 8.**

# Appendix A

**Table A1. Characteristics of the stations.**

| Number | Name | Latitude | Longitude | Altitude | Model level |
|:---:|:---:|:---:|:---:|:---:|:---:|
| 1 | Abresan (Tabriz) | 38.066 | 46.326 | 1440 | 137 |
| 2 | Namaz square (Tabriz) | 38.079 | 46.289 | 1393 | 137 |
| 3 | Azarbayejan square (Tabriz) | 38.112 | 46.276 | 1362 | 137 |
| 4 | Aqdasiyeh (Tehran) | 35.795 | 51.484 | 1528 | 123 |
| 5 | Sadr (Tehran) | 35.778 | 51.429 | 1488 | 124 |
| 6 | District 2 (Tehran) | 35.777 | 51.368 | 1559 | 123 |
| 7 | Punak (Tehran) | 35.762 | 51.332 | 1468 | 124 |
| 8 | Geophysics (Tehran) | 35.74 | 51.385 | 1368 | 126 |
| 9 | Setad bohran (Tehran) | 35.727 | 51.431 | 1284 | 129 |
| 10 | Tarbiat Modares (Tehran) | 35.717 | 51.386 | 1264 | 129 |
| 11 | Sharif university (Tehran) | 35.702 | 51.351 | 1187 | 132 |
| 12 | District 21 (Tehran) | 35.698 | 51.243 | 1219 | 131 |







| 13 | Piroozi (Tehran) | 35.696 | 51.494 | 1209 | 131 |
| 14 | Fath square | 35.679 | 51.337 | 1151 | 133 |
| 15 | Shad abad (Tehran) | 35.67 | 51.297 | 1157 | 133 |
| 16 | Mahallati (Tehran) | 35.661 | 51.466 | 1139 | 134 |
| 17 | District 19 (Tehran) | 35.635 | 51.362 | 1103 | 135 |
| 18 | Masoudieh (Tehran) | 35.63 | 51.499 | 1177 | 136 |
| 19 | Ray | 35.604 | 51.426 | 1065 | 137 |


**Table A2. The standard deviation for the S component of the variables at station 8.**

| Variables | $O_3$ | $NO_x$ | AT | WS | WD |
|---|---|---|---|---|---|
| dfo | 18.11 | 43.36 | 2.76 | 0.87 | 42.77 |
| dfr | 44.93 | 82.33 | 2.69 | 0.55 | 66.07 |
| dfa | 19.09 | 392.44 | 2.88 | 0.69 | 79.76 |


**Table A3. The standard deviation for the M component of the variables at station 8.**

| Variables | $O_3$ | $NO_x$ | AT | WS | WD |
|---|---|---|---|---|---|
| dfo | 4.88 | 20.89 | 1.59 | 0.38 | 21.84 |
| dfr | 6.83 | 28.39 | 1.43 | 0.36 | 44.91 |
| dfa | 2.5 | 202.41 | 1.43 | 0.41 | 60.16 |




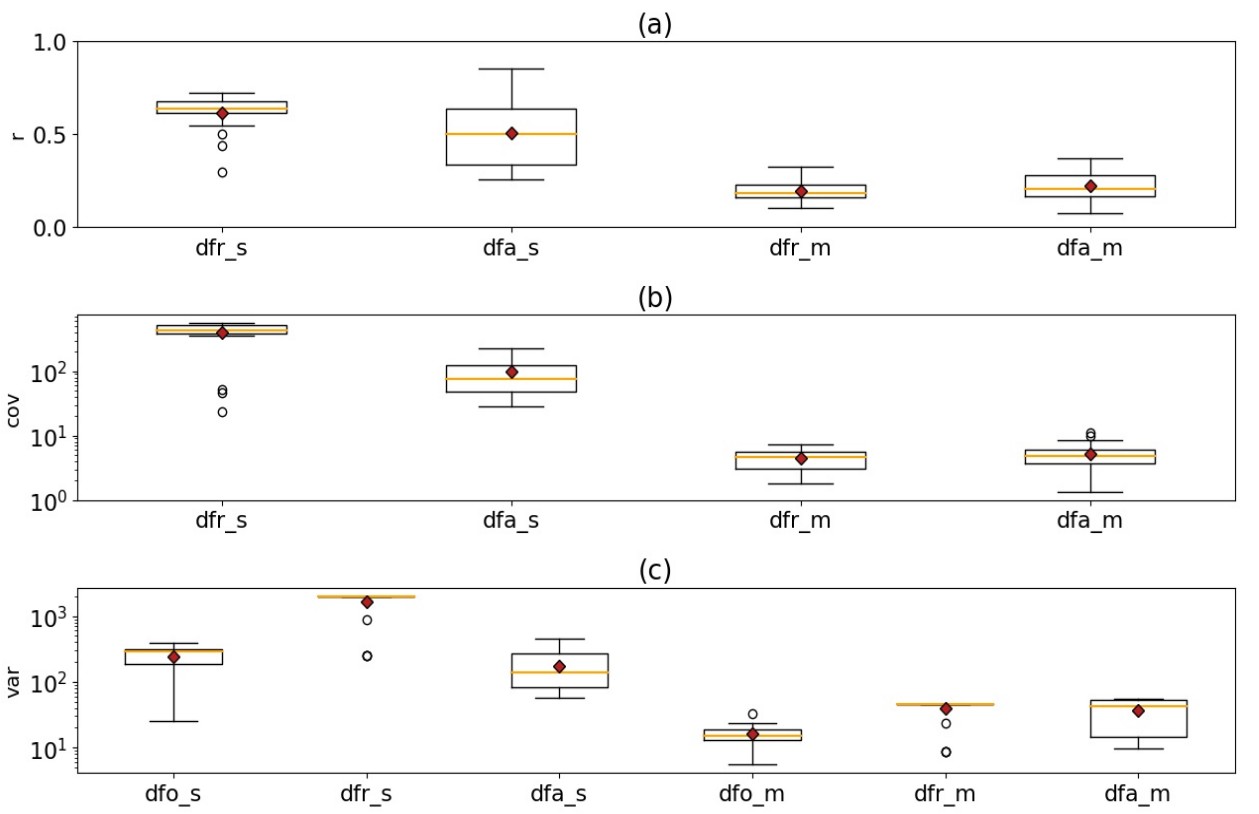

**Figure A1.** The (a) correlation, (b) covariance, and (c) variance of the simulated $O_3$ in dfr, and dfa for both S and M components.

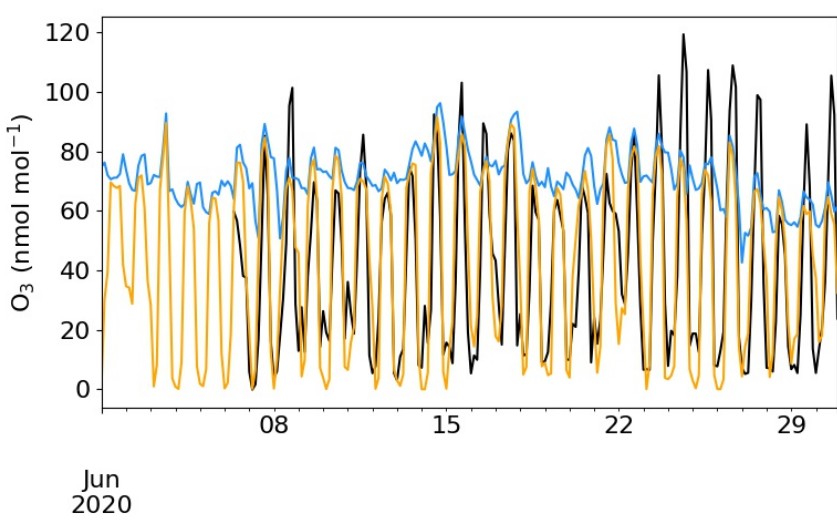


**Figure A2.** The $O_3$ time series for June 2020 at station 4. The black line shows observations (dfo) and orange and blue colours show dfa in two different model levels, i.e., 137 and 123, respectively.





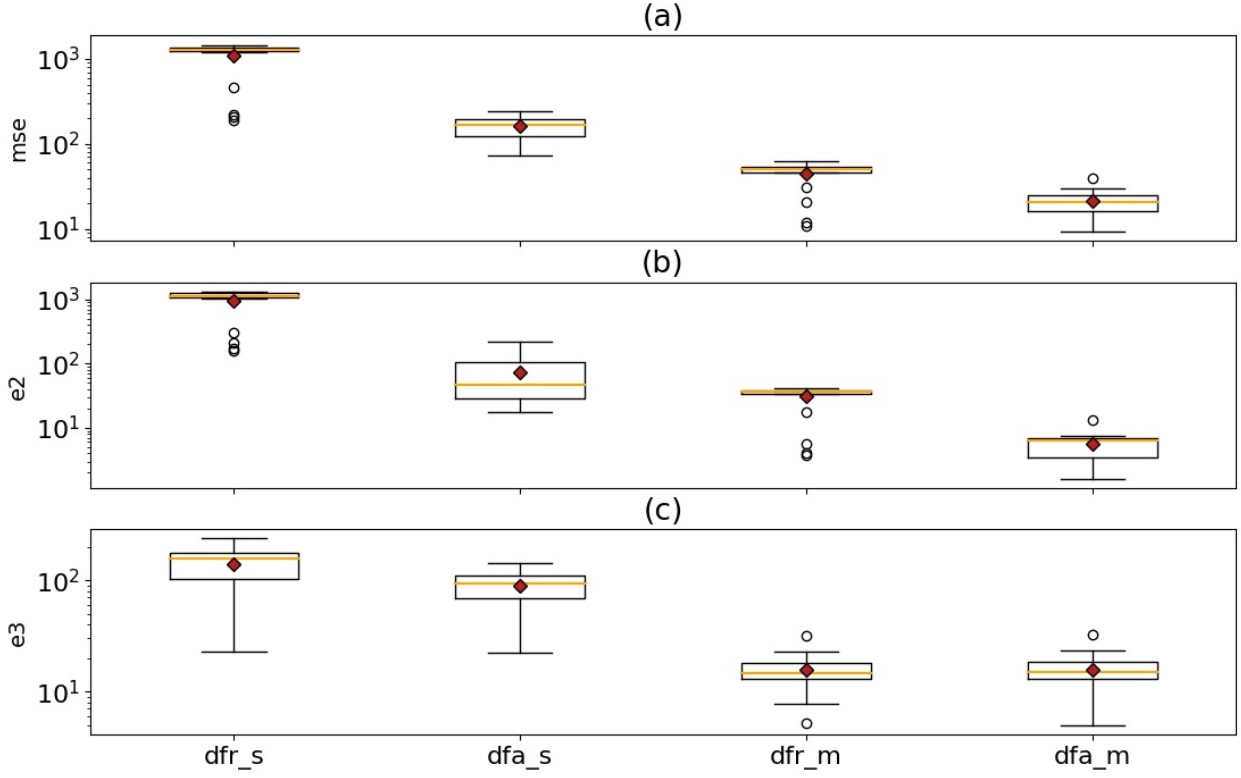

**Figure A3. Similar to Fig. 2, but here the surface level is used as a vertical model level, i.e., lev = 60 for dfr and lev = 137 for dfa.**






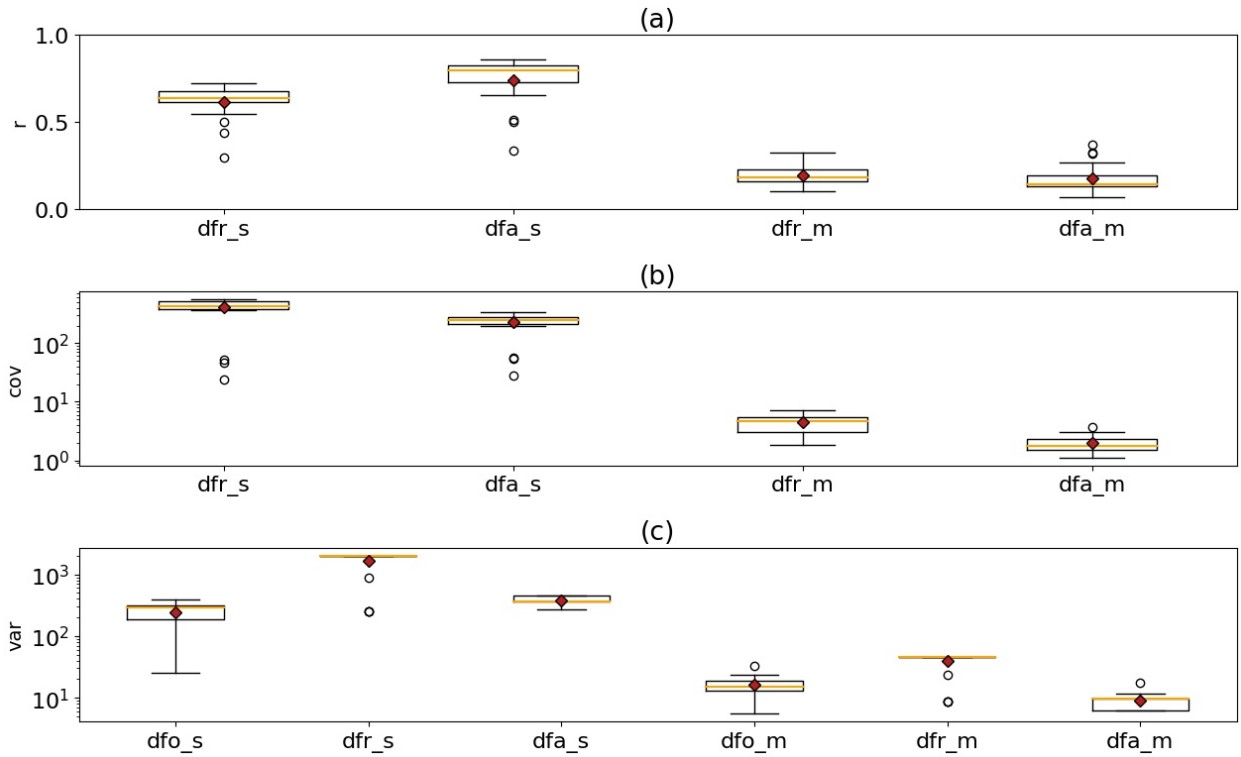

**Figure A4. Similar to Fig. A1, but here the surface level is used as a vertical model level, i.e., lev = 60 for dfr and lev = 137 for dfa.**