# Peer review of "Assessment of tropospheric ozone products from CAMS reanalysis and near-real time analysis using observations over Iran"

_Geoscientific Model Development, 2022_

## Author Comment (AC1)

The authors acknowledge the referee for spending time and providing valuable and punctilious comments. Although most of the suggestions were applied in the revised paper, here are a few points to be mentioned:

The study assesses the modeled surface ozone variability over two parts, north and northwestern, of Iran. Two CAMS products, namely reanalysis and daily forecast, were evaluated against surface-based measurements at the city of Tehran (a megacity) and Tabriz (an industrialized city).

Please note that in this document, the tables and figures are referred based on their numbers in the revised article.

- Although most studies (prefer to) evaluate global models versus observations at the background (rural) sites, neglecting urban stations leads to a biased evaluation, in particular where a large fraction of the grid box is associated with urban conditions, e.g., megacities. Assuming a single grid box over the cities, that can give some relevant information about ozone variability in those regions. The regional nature of ozone, with a space scale up to 500 km, has been shown over different parts of the globe. Apart from it, using several stations distributed in a grid box provide more reliable results than the case of using one station per grid box; because the results can be affected depending on the position of the station, i.e., close to a grid point, or away from it. Most of the air quality monitoring stations in Iran are installed in the cities, as they are aimed for the public health report. There is no information about stations' type or availability of the data at background sites. The Geophysics station is located at the Geophysics institute, University of Tehran, Tehran. This station measures surface ozone, along with several other variables such as air temperature, nitrogen oxides, wind, total ozone column, etc. These data are often used for research studies.
- Global models with coarse resolutions might have deficiencies to describe the process at local scales, but the current generation of reanalysis datasets or global models with fine resolutions are expected to capture some of those processes (at least to some extent). The altitudes of the stations are larger than 1000 m, which might not be resolved by the models. So, the results of the reanalysis datasets are based on the surface model level (= 60). For forecast datasets, the optimum vertical model levels are shown in Table A1. In addition, the results at the surface model (= 137) of forecast datasets are discussed in Sect. 5.
 - That was a great idea and described in the (revised) paper. The initial thought of the authors was to refer the reader to the other publications, but since each study uses a given version of the models, that was best to describe these aspects.
- Data series of ozone and several proxies are taken for all datasets, i.e., observations and both models. The predictors were chosen based on data availabilities and literature, in which these quantities are often used as proxies in explaining ozone variabilities. Nevertheless, there are studies in which other quantities such as radiation and cloud cover are used as ozone predictors. The first point to be mentioned is that there is a significant correlation

between temperature and radiations, so using both quantities may not add (or explain) more information. That can be tested using AIC, BIC, and lasso. Second, the point of this analysis is to compare the relationship between these quantities and ozone in reality and simulations, so the data needs to be available for three datasets. For easier comparisons of the results (both components of three datasets), they are shown in one figure (Fig. 4) instead of listing in a table.

The suggestions were to the point. The mentioned plots were meant to show the components over a given time period, but the suggestion is right. Therefore, the plot (Fig. 2) was created upon the referee's opinion. The naming conventions had arisen from the programming style of the authors, where 'df' refers to the data frame. Table 1 shows the coefficients for standardized datasets to show the relative importance of the variables. They only depend on the standard deviations, which are given in table A2 and table A3. The coefficients for non-standardized data are shown in table A4. To have a co-located data series of ozone and predictors, we selected datasets of station 8. Although model observation comparisons based on aggregated data and associative metrics provide information about the model performance; they often describe major features, and do not target the sources of error. Here we present an approach to compare observation and models that focus on given timescales. Scale analysis helps to identify the possible sources of errors, and underlying physical or chemical processes in each model. Metric used in the study helps to find the nature of the errors (bias, model-related, or observation-related error), which can be a diagnostic tool helping developers to improve the models. We used a regression model to infer the relevant processes which could have generated the model errors. Furthermore, it can be used to understand how the relationship between ozone and its proxies changes on various timescales.

---

## Author Comment (AC2)

The authors thank the referee for spending time and providing valuable and punctilious comments. Although most of the suggestions were applied in the revised paper, here are a few points to be mentioned:

The study presents an approach for a detailed assessment of two CAMS products, namely reanalysis (CAMSRA) and daily forecast (CAMSFC), over two parts, north and northwestern, Iran.

Please note that in this document, the tables and figures are referred upon their numbers in the revised article.

- That is right, the study is regional as (surface) ozone is a regional issue. Iran is a country with a complex topography and diverse meteorological systems. Exposure to the high concentration of air pollution, especially ozone, leads to premature deaths, in particular, those suffering from asthma disease. Iran is not an exception to that as hundreds of premature deaths are attributed to ozone in Tehran within a year. It has been shown that over Tehran the ozone levels are controlled by local emissions and several (given) synoptic systems. Air pollution is also one of the major environmental issues in Tabriz, a metropolitan area in northwestern Iran. Polluting industries such as thermal power plants and oil refineries in the west of the city are accounted for poor air quality over Tabriz. This city surrounds by mountains and is located in the vicinity of the eastern Mediterranean. Various factors in different timescales affect the ozone over these cities, which brings them to be a hot spot region to study ozone and show the performances of CAMSRA and CAMSFC in simulating ozone over these areas. That is a great idea to expand the study over different regions such as the Middle East or Europe. But one has to take into account limitations on the measured data availability, at least in the Iran neighborhoods. The study can be done in Europe, where more data are available. That has to be carried out in a separate study (either by us or other scientists) as ozone characteristics and their associations differ across regions. For instance, stratospheric-tropospheric intrusions play more role in tropospheric ozone variability over the eastern of the Mediterranean than the western part, where the lack of strong synoptic advection combined with the orographic characteristics and the sea-land breezes favor episodes of high ozone levels over this region. So, for the detailed assessments, it is preferable to carry the study per region.

- Various physical and chemical processes can affect ozone variabilities but at different timescales. The S variations, which are composed of intraday and diurnal motions, are more attributed to the local (photo)chemistry and daytime-nighttime chemistry. The process within the day or in short time scales (e.g., titration, vertical mixing, etc.) or diurnal variation of solar flux can affect the S variability. Concerning emissions or deposition, they can affect ozone on various time scales, i.e., short, seasonal, and longer time scales; For instance, deposition can act slowly due to the change in surface properties, such as lead area index, drought conditions, long-term erosions, harvesting, etc. Local source emissions influence short-scale

ozone variabilities. Stratospheric-tropospheric transfer can be considered as a seasonal event, as monsoon controls the seasonality of fold occurrence and its intensity over the whole region of the Eastern Mediterranean and the Middle East. That has little insight into S variability.

- Suggestion was great and details were provided accordingly in the revised paper.

- The MLR was applied for all three datasets. The results for the observation datasets are listed in a table, and the results of all datasets have been shown in Fig. 4 for easier comparison.

- That was not well written, as it was meant to be "bias of surface ozone is larger than free tropospheric ozone".